# What Is the Link of Closeness and Jealousy in Romantic Relationships?

**DOI:** 10.3390/bs15020132

**Published:** 2025-01-26

**Authors:** Ana María Fernández, Maria Teresa Barbato, Pamela Barone, Belén Zavalla, Diana Rivera-Ottenberger, Mónica Guzmán-González

**Affiliations:** 1Laboratorio de Evolución y Relaciones Interpersonales, Universidad de Santiago de Chile, Santiago 9170022, Chile; maria.barbato@usach.cl (M.T.B.); belen.cordero@usach.cl (B.Z.); 2Department of Psychology, Universitat de les Illes Balears, 07122 Palma, Spain; pamela.barone@uib.es; 3School of Psychology, Pontificia Universidad Católica de Chile, Santiago 7820436, Chile; dvrivera@uc.cl; 4School of Psychology, Universidad Católica del Norte, Antofagasta 1270709, Chile; moguzman@ucn.cl

**Keywords:** close relationships, romantic bonds, emotions

## Abstract

From an evolutionary perspective, love and attachment foster closeness, while jealousy ensures exclusivity in romantic relationships. This study examined the links between jealousy and affective aspects of love, hypothesizing positive associations despite their apparent opposition. An online sample of 265 individuals in Chile and Spain completed measures of digital jealousy, closeness, love, felt loved, and attachment. Results revealed higher jealousy in Chile than in Spain. Across both countries, anxious attachment and closeness were significant predictors of jealousy, explaining nearly 30% of its variance. In Chile, feeling loved negatively predicted jealousy, suggesting that reassurance of the romantic bond may reduce jealousy in this cultural context. Notably, affective closeness—conceptualized as the inclusion of the self in the other—emerged as a novel predictor of jealousy, extending beyond the established role of anxious attachment. These findings underscore the nuanced interplay between cultural context, affective closeness, and attachment in shaping jealousy.

## 1. Introduction

### 1.1. Love and Attachment

Love is a complex and multifaceted emotion that plays a crucial role in promoting enduring social bonds, especially within romantic relationships ([43]). From an evolutionary perspective, romantic love is a mechanism that encourages long-term commitment, facilitating caregiving and protection of offspring, both of which are essential for survival and reproductive success ([15]; [71]). Love extends beyond romantic partnerships, influencing family and friendship dynamics, where consistent emotional closeness and actions that convey love are essential ([79]). In this context, the emotional connection one shares with others is crucial for feeling loved, not just within romantic relationships but across various forms of social bonds.

The theory of attachment supports this understanding by emphasizing that early relational experiences with caregivers shape mental models that influence adult relationships ([56]). These models underline how trust, emotional security, and mutual support within relationships foster emotional closeness, making such connections indispensable ([6]; [44]). Attachment bonds are fundamental not only in parent–child relationships but also in adult romantic relationships, reinforcing that emotional security and attachment are key to sustaining love over time ([28]; [29]). Yet, attachment theory is just one lens through which love can be understood. In addition, [36]’s ([36]) theory of “positivity resonance” broadens the view of love, proposing it as a dynamic, momentary connection rooted in shared positive emotions, biobehavioral synchrony, and mutual care. This theory aligns with the idea that emotional closeness integrates a partner into one’s sense of self, strengthening the bond ([1]). Secure attachment further reinforces positive emotional resonance, supporting the notion that love involves a complex interplay of emotional security and mutual care ([56]).

Despite the wide range of components that characterize love, the most commonly utilized framework for its measurement is Sternberg’s Triangular Theory of Love. This theory portrays love in three dimensions: intimacy, passion, and commitment, which can be ranked to convey different types of love ([73]). Intimacy refers to feelings of closeness and emotional connection, passion relates to physical attraction and desire, and commitment involves the decision to maintain a relationship over time. Stenberg’s model has significantly advanced the characterization of romantic relationships by providing a comprehensive structure to understand the dynamics of love cross-culturally ([16]; [49]).

Moreover, the dimension of intimacy emphasizes that feelings of being loved are deeply rooted in mutual understanding and responsiveness between partners. This dimension can be measured through aspects of momentary connection, fostering deeper bonds and empathy within the relationship ([36]; [38]; [60]).

However, there is a complementary perspective on the nature of love, emphasizing the role of neuropsychological and emotional responses in shaping romantic experiences. For example, the concept of “feeling loved” highlights individual perceptions of love based on momentary emotional interactions, which may not always align with the structural components of intimacy, passion, and commitment ([30]; [45]; [60]). The dynamics of love appear to be shaped not only by long-term relational structures but also by transient, emergent experiences of closeness and affection ([62]).

Recent research underscores the importance of measuring the subjective experience of “feeling loved” as a critical dimension in understanding relationships ([65]). This perspective enriches relational bonds by emphasizing the nuanced, reciprocal, and momentary aspects of love. Integrating this measurement with established theories deepens our understanding of how love operates across various relational contexts.

On the other hand, several studies have revealed that the way romantic love is experienced and expressed varies significantly across cultures and is shaped by societal norms and traditions ([48]). In Western cultures, love is often expressed explicitly through verbal affirmations and grand gestures, while in Chinese and Filipino traditions, it is more commonly conveyed through implicit actions and subtle behaviors ([58]). Emotional investment also shows cultural differences, with North Americans tending to express higher levels of affection and passion compared to the more reserved approaches typical in East Asia ([67]). Moreover, cultural attitudes toward love differ, as more individualistic societies, such as Spain compared to Chile ([47]) (https://www.hofstede-insights.com/), emphasize autonomy and passionate connections, whereas collectivistic cultures prioritize familial bonds and societal harmony ([27]).

Across the majority of these studies, the extensive use of instruments enhances the ability to assess phenomena effectively, yet it remains unclear what specific aspects or characteristics of love are being evaluated. For instance, some cross-cultural studies on love employ standardized tools such as emotional investment scales to assess levels of affection, passion, and compassion across cultures ([67]). Similarly, [25] ([25]) examined dimensions like altruism and physical attraction using self-report measures, while [72] ([72]) explored cross-cultural differences in the intensity of love through direct surveys. Longitudinal studies, such as those by [46] ([46]), have analyzed the evolution of romantic relationships over time, capturing shifts from initial intimacy to later emotional development in various cultural contexts. These tools, combined with ethnographic approaches, provide valuable insights into how universal emotions like love are shaped by cultural norms, revealing both shared and unique elements across societies.

Love not only fosters emotional security, mutual care, and intimacy but also evokes a range of powerful emotions that differentiate and define human relationships, from infancy through adulthood ([5]; [49]). Among them, jealousy stands out as a pivotal emotion, intricately tied to the dynamics of close relationships and deeply influencing how love is experienced and expressed ([75]).

### 1.2. Jealousy

Jealousy represents a pivotal emotion in close human relationships, primarily examined in children within the framework of attachment theory ([6]; [33]; [42]), as well as in the contexts of friendships and sibling dynamics ([61]; [56]). Although often characterized as a troublesome emotion ([61]), jealousy also plays a complementary role to love, serving to preserve romantic bonds ([34]). From an evolutionary perspective, jealousy functions as a defense mechanism aimed at safeguarding relationships from perceived threats ([18]), particularly infidelity or emotional disengagement ([8]; [68]). Commonly regarded as a reaction to the fear of losing a partner’s affection ([12]), jealousy fulfills an adaptive function by protecting the emotional and reproductive resources essential for sustaining long-term relationships ([9]). In this regard, love and jealousy operate synergistically to stabilize romantic partnerships ([22]; [35]).

Research on jealousy has concentrated on its evolutionary and functional aspects, while also exploring the cultural and sexual variations that influence its expression. Cultural dimensions, such as collectivism and individualism, have been shown to influence how jealousy is experienced. [80] ([80]) found that collectivist societies prioritize relational harmony, often perceiving jealousy in the context of maintaining group stability, whereas individualistic cultures emphasize personal autonomy, framing jealousy as a threat to individual relationships. These cultural differences have been measured using tools like the Horizontal and Vertical Individualism and Collectivism Scale ([70]), which provides valuable insights into the broader social frameworks that modulate emotional responses to jealousy.

In the context of romantic relationships, jealousy has been identified as a crucial emotion for protecting mating bonds ([14]; [31]). It has primarily been studied through hypothetical infidelity scenarios and retrospective accounts of distress related to actual infidelity ([11]; [9]; [68]). For example, men tend to exhibit greater distress by sexual infidelity, aligning with evolutionary concerns about paternity certainty, while women report heightened emotional jealousy, often linked to perceived threats to emotional commitment ([11]; [13]; [53]). These patterns have been captured using tools like the modified Emotional and Sexual Jealousy Scale ([11]) and the Self-Report Jealousy Scale ([7]), which quantify reactions to emotional and sexual threats. Experimental methods, including recalling personal betrayal experiences, exposure to attractive rivals, dramatized depictions in media (e.g., Cheaters), and subliminal priming with infidelity-related cues, have further highlighted the nuanced ways in which jealousy is elicited and expressed ([51]; [54]; [63]).

Jealousy poses a significant threat to intimate relationships, as it is often considered a maladaptive and pathological emotion rooted in insecurity and negative self-esteem ([17]; [10]). Previous research has linked jealousy to traits like narcissism, psychopathy, Machiavellianism, and low self-worth ([24]; [55]; [69]). Despite its association with psychological distress and interpersonal conflict, jealousy remains universally prevalent, even in cultural contexts where traditional explanations—such as male dominance, aggression, or female emotional vulnerability—fail to fully explain its persistence and intensity ([50]; [66]). This paradox highlights the complexity of jealousy as an emotional phenomenon.

Qualitative approaches have complemented these findings by uncovering deeper themes, such as infidelity, expectations of time and commitment, self-esteem, and the influence of social media ([80]). Additionally, innovative methods like economic games, which manipulate resource allocation between romantic partners and rivals, have provided new ways to study jealousy by offering real-time insights into emotional responses ([3]). The rise of social media has introduced another layer, amplifying relational threats and rivalries in the digital sphere, and making it an increasingly important area of study ([76]; [77]).

### 1.3. Relationship Between Love and Jealousy

Therefore, both love and jealousy play complementary roles in promoting and preserving pair bonds ([18]). Love fosters prosocial behaviors and mutual benefits between reproductive partners ([35]), while jealousy helps prevent the loss of these benefits in the face of potential interlopers ([9]; [41]). These emotions together support the maintenance of long-term commitments, contributing to the evolutionary success of pair bonding by ensuring the continuity of shared resources, care, and reproductive fitness ([22]; [23]).

The relationship between love and jealousy is influenced by various factors such as attachment style ([2]; [4]), given the convergence of both constructs in the development of close bonds ([34]). Evidence suggests that high attachment anxiety is strongly associated with increased jealousy ([75]). More specifically, individuals with anxious attachment often experience heightened suspicion, insecurity about their partner’s availability, and concerns about potential abandonment, all of which can trigger jealousy ([21]; [26]; [40]; [57]; [59]). In contrast, attachment avoidance—a tendency toward emotional independence due to early neglect or a lack of sensitive care—leads to self-reliance and discomfort with affection, often resulting in reduced jealousy ([52]; [57]).

Therefore, the present research aims to explore the interplay between jealousy, emotional closeness, and love across two distinct cultural contexts: Chile and Spain. Drawing on attachment theory and theories of romantic love, this study examines how jealousy relates to key relational variables that influence the stability and dynamics of romantic relationships. Specifically, it is hypothesized that jealousy will show positive associations with emotional closeness, love (including its dimensions of intimacy, passion, and commitment), and attachment anxiety, as these variables are intertwined in shaping individuals’ emotional responses. By exploring these dynamics, this study seeks to shed light on the role jealousy plays within the broader context of love, and how attachment, emotional closeness, and cultural factors shape the experience and expression of jealousy in romantic relationships.

## 2. Materials and Methods

### 2.1. Participants

This study employed a cross-sectional design with a convenience sample of 265 romantically involved individuals who completed all measures online using the PsyToolkit platform ([74]). Participants were recruited through advertisements on social networks, resulting in a sample composed predominantly of heterosexual individuals (74.3%), of whom 71.9% were women. Only 29.2% of participants reported having children.

The sample included participants from Chile (40%) and Spain (60%). In the Chilean subsample, 78% were women compared to 72% in the Spanish subsample. There was a significantly shorter relationship duration in months among participants from Chile (*M* = 46.97, *SD* = 6.46) compared to those from Spain (*M* = 89.18, *SD* = 15.63; *t*_231.255_ = −2.494, *p* < 0.05). No significant differences were found between the two countries in terms of age or sex across any of the assessed variables (*F*s < 1.43, *p*s > 0.05).

### 2.2. Measures

Sociodemographic questions. Participants provided information about their age, sex, employment status, duration of their current relationship, and number of children.

Jealousy. The digital jealousy scale (DJS) was used to measure jealousy in digital contexts ([39]). This 9-item scale covers cognitive, affective, and behavioral aspects of jealousy. Participants were instructed to think about their current relationship and indicate their level of agreement with each statement on a 6-point Likert scale (1 = strongly disagree; 6 = strongly agree). Example items include “it worries me when a new woman/man appears on my partner’s friends list” or “I look over my partner’s shoulder when I know they are texting with someone else”.

The original scale was developed in both German and English. For the current study, the items were translated into Spanish from the English version by a bilingual team, and a back-translation process was employed to ensure accuracy. In the original study, the DJS showed high construct reliability (McDonald’s ω = 0.89 in the German sample and McDonald’s ω = 0.90 in the English sample). Cronbach’s alpha for the adapted DJS was 0.90. [39] ([39]) found that digital jealousy and romantic jealousy (measured by a scale that does not refer to the context of social media) showed a very strong positive correlation (r = 0.86).

Love. The short version of the Triangular Love Scale (TLS-15) was used to measure love ([49]). Based on Sternberg’s triangular theory of love, the scale evaluates three core components, intimacy, passion, and commitment, commonly measured in romantic relationships. The abbreviated version consists of 15 items rated on a 5-point Likert scale (1 = never; 5 = always). Example items of each subscale include “I have a warm relationship with my partner” (intimacy), “I find my partner to be very personally attractive” (passion), and “I view my commitment to my partner as a solid one” (commitment).

The Spanish version of the TLS-15 has been previously validated and adapted for both Latin American and Spanish populations, showing high reliability in both contexts. In the Latin American sample, the reliability scores were intimacy: α = 0.92, ω = 0.94; passion: α = 0.89, ω = 0.91; and commitment: α = 0.90, ω = 0.93. In the Spanish sample, the scores were intimacy: α = 0.91, ω = 0.93; passion: α = 0.88, ω = 0.90; and commitment: α = 0.90, ω = 0.90 ([49]). For the present sample, the reliability scores for the TLS-15 were the following: intimacy: α = 0.86; passion: α = 0.81; and commitment: α = 0.83.

Closeness. An adapted version of the inclusion of other in the self (IOS) scale ([1]), a valid measure of perceived closeness between individuals ([37]), was used. This single-item scale employs visual representations of overlapping circles to illustrate varying degrees of identification with the partner and affective closeness. Participants are asked to select the pair of circles that best represents their relationship with their romantic partner.

Felt loved. The Felt-Loved scale was used to assess how loved participants felt about their partners ([65]). Participants rated the following three items on a 7-point Likert scale (1 = not at all; 7 = very much): “I feel cared for/loved by my partner”; “I feel accepted/valued by my partner”; and “I feel understood/validated by my partner.” The original study averaged the items to create an overall measure of felt love (α = 0.94; Sasaki et al., 2023). Cronbach’s alpha for the current sample was 0.84.

Attachment. The Revised Adult Attachment Scale (RAAS), Close Relationship version ([20]), was used to assess adult attachment styles. This 18-item scale measures emotional closeness or distance in relationships using a 5-point Likert scale (1 = not at all characteristic of me; 5 = very characteristic of me). It evaluates three dimensions of attachment: closeness, dependence, and attachment-related anxiety and avoidance. Example items of each subscale are “I find it relatively easy to get close to people” (close); “I find it difficult to allow myself to depend on others” (depend); and “I often worry that other people don’t really love me” (attachment-related anxiety and avoidance)

The validated Spanish version of the scale was used ([32]). Only the dimensions of anxiety (Cronbach’s alpha = 0.90) and avoidance (Cronbach’s alpha = 0.81) are reported in the current study.

### 2.3. Procedure

Data collection occurred between April and July 2024 using an online platform, where participants completed the questionnaires mentioned above. Following APA ethical standards, informed consent was obtained from all participants before participation. The study was approved by the Institutional Ethics Committee of the first author’s university, ensuring adherence to confidentiality and ethical guidelines throughout the research process.

### 2.4. Data Analysis

Descriptive statistics were calculated to summarize the characteristics of the sample. Mean differences across countries were examined using independent t-tests, and effect sizes were calculated using Cohen’s d.

Pearson correlations were performed to explore associations among the affective variables. Additionally, stepwise multiple regression analyses were conducted to identify the best predictors of jealousy. Separate regressions were conducted for the overall sample and by country (Chile and Spain) to examine potential cultural differences in the predictors of jealousy. All analyses were performed using SPSS 25 and JAMOVI 2.4.5. with significance levels set at *p* < 0.05.

## 3. Results

All participants in the sample were in a committed relationship when they completed the scales. Additionally, they reported having had on average 2.21 romantic relationships (*SD* = 1.24), having an average of 0.56 children (*SD* = 0.99).

The normality of the data was assessed using the Shapiro–Wilk test. The results indicated that all variables, including jealousy (*W* = 0.165, *p* = 0.000), IOS (*W* = 0.165, *p* = 0.000), intimacy (*W* = 0.193, *p* = 0.000), passion (*W* = 0.207, *p* = 0.000), commitment (*W* = 0.113, *p* = 0.000), felt loved (*W* = 208, *p* = 0.000), and anxious attachment (*W* = 0.125, *p* = 0.000), did not meet the normality criterion, while avoidant attachment (*W* = 0.053, *p* = 0.079) was normally distributed.

Descriptive statistics by country are presented in Table 1, with Chilean participants reporting significantly more jealousy than the Spanish (*t* = 3.653, *p* < 0.001, Cohen’s *d* = 0.47).

When inquiring about the only demographic variable that was different across countries, the correlation between time in the relationship and the affective variables yielded that, in the Chileans, this was directly correlated to age (*r* = 0.61, *p* < 0.001), and inversely associated with jealousy (*r* = −0.31, *p* < 0.01) and anxious (r = −0.38, *p* < 0.01) and avoidant attachment (*r* = −0.23, *p* < 0.01). In the Spanish participants, only age, again, was positively correlated with time in the relationship (*r* = 0.41, *p* < 0.001).

Correlational analyses in the whole sample among the affective variables showed that jealousy was significantly and directly associated with IOS and anxious and avoidant attachment and inversely correlated with the dimensions of intimacy and commitment from the Love Scale, as well as the Felt-Loved scale (See Table 2).

A stepwise regression predicting jealousy from the affective variables yielded anxious attachment, IOS, and felt loved as the best predictors of jealousy (R^2^ = 0.28, *F*_3, 254_ = 33.264. *p* < 001). Jealousy was positively predicted by anxious attachment (β = 0.325. *t* = 5.772, *p* < 0.001) and IOS (β = 0.279, *t* = 5.082, *p* < 0.001) and negatively predicted by felt loved (β = −0.176, *t* = −2.910, *p* < 0.001).

As can be seen in Figure 1, two additional stepwise regressions were run predicting jealousy by country. In Chile, 41.9% of the variance in jealousy (*F*_3, 89_ = 20.650, *p* < 001) was positively explained by IOS (β = 0.395. *t* = 4.433, *p* < 0.001) and anxious attachment (β = 0.218, *t* = 2.364, *p* < 0.05) and negatively predicted by felt loved (β = −0.251, *t* = −2.827, *p* < 0.05). In Spain, 21.9% of the variance in jealousy (*F*_2, 165_ = 23.157, *p* < 0.001) was predicted by anxious attachment (β = 0.400, *t* = 5.749, *p* < 0.001) and IOS (β = 0.188, *t* = 2.705, *p* < 0.01).

## 4. Discussion

This study posits that love and jealousy are emotions that help protect a significant romantic relationship, rather than being reverse affective variables, as might commonly be expected. Therefore, jealousy was predicted to be associated with relationship closeness and insecure attachment, a hypothesis that was supported by our results from two Spanish-speaking samples, one from South America and the other from Europe.

The results reveal distinct patterns in how demographic and affective variables interact across the two countries. For Chilean participants, relationship duration correlates significantly with a broader range of variables, including age, jealousy, and attachment styles. This suggests a more complex interplay between relationship duration and emotional dynamics in Chile. Longer relationships may foster lower attachment insecurity and jealousy, which aligns with the classical literature that links jealousy to insecure attachment ([61]). Additionally, shorter relationship duration may signal lower commitment. In contrast, among Spanish participants, relationship duration was found to correlate exclusively with age. This possibly reflects a distinct cultural approach to relationship dynamics, where the passage of time exerts less influence on affective experiences. Furthermore, the absence of a significant association between relationship duration and jealousy in the Spanish sample may be explained by a more individualistic orientation in the Spanish sample than in the Chilean group. It could also be linked to a higher prevalence of avoidant attachment styles in more individualistic societies ([64]). Supporting this interpretation, [26] ([26]) reported that individuals with avoidant tendencies tend to experience less pervasive jealousy in Russia.

Jealousy is consistently associated with anxious attachment and the perception of the inclusion of other in the self (IOS) across both samples, highlighting the universal role of attachment insecurity and relational closeness in this affection. However, the perception of being loved (felt loved) emerged as a protective factor against jealousy, particularly in the Chilean sample. This outcome suggests that feelings of relational security may mitigate jealousy more strongly in Chile, reflecting cultural nuances in emotional regulation or relational expectations ([21]), which could be related to the shorter relationship duration of this group compared to the Spanish. For example, [65] ([65]) found that feeling loved fosters emotional security and mutual appreciation within a couple, serving as a protective factor for relationship stability. This variable may act as a buffer against negative emotions, reducing their potential to disrupt relational harmony. By reinforcing positive affect and strengthening emotional bonds, feeling loved helps couples navigate challenges more effectively, reducing conflict escalation, enhancing resilience, and mitigating romantic jealousy. Alternatively, the shorter relationship duration observed in the Chilean sample could be associated with an enhanced sensibility to jealousy, as limited romantic stability is known to increase jealousy ([19]).

The variance in jealousy explained by the regression models differs notably between countries, with the Chilean data explaining nearly twice the variance observed in Spain. This disparity underscores the potentially greater emotional complexity or cultural significance of jealousy in Chilean relationships. The stronger role of feeling loved among Chilean participants highlights the protective influence of emotional validation and perceived affection, which seems less critical in the Spanish sample (see [78]).

The findings suggest that jealousy, rather than being merely an indicator of insecurity or distrust, may reflect the emotional intensity characteristic of close relationships. In the Chilean sample, the perception of feeling loved served as a significant protective factor, stressing the role of relational security in mitigating jealousy. Greater emotional closeness appears to moderate jealousy, as reciprocal connections within relationships enable individuals to manage these emotions more effectively ([2]; [65]).

The difference in the impact of feeling loved between Chile and Spain may stem from the varying emphasis on emotional connection within these cultural contexts. Individualistic societies often prioritize autonomy and passionate connections, while collectivist cultures place greater value on family ties and social harmony ([58]). These cultural differences extend to responses to jealousy, particularly in situations involving sexual infidelity, where factors such as commitment expectations and the influence of social media play significant roles ([80]). In Chile, shorter average relationship durations may reflect distinct relational expectations shaped by cultural or generational factors. For instance, younger generations or individuals from collectivist cultures may approach relationship maintenance behaviors with heightened sensitivity to emotional security and perceived trustworthiness. Previous research suggests that relational expectations evolve with cultural norms and age groups, influencing thresholds for acceptable behavior on social media ([76]). These evolving norms could explain why shorter relationships in Chile exhibit greater sensitivity to feeling loved, compared to the longer and potentially more stable relationships observed in Spain.

The findings point to the importance of considering cultural context when examining jealousy and its predictors. In Chile, jealousy seems deeply intertwined with affective security and emotional closeness, whereas, in Spain, attachment anxiety and perceived relational closeness (IOS) play dominant roles, with a weaker association with feeling loved. These distinctions may reflect cultural differences in attachment processes, relational priorities, or societal norms surrounding romantic relationships. For example, [64] ([64]) found that there is an association between attachment anxiety and interdependence—the more individuals view themselves in relation to others, the more attachment anxiety they experience—which may underlie greater feelings of jealousy in the Chilean context since there is higher relational closeness and possibly interdependence.

While this study illuminates key predictors of jealousy, the cross-sectional nature, small sample size, and convenience sampling limit causal interpretations. Additionally, jealousy was measured using a digital jealousy scale, which may not fully capture the broader context of jealousy in romantic relationships. Another limitation of this study was that the distribution of sexes within the sample was not balanced, which could influence the generalizability of the findings. Furthermore, as most participants were in heterosexual relationships, we were unable to analyze potential differences in affective variables based on relationship type, like in polyamorous or fluid relationships.

Future research should explore longitudinal designs to track changes in affective variables over time and examine how cultural factors shape emotional regulation in relationships. Additionally, investigating other cultural contexts would further elucidate the universality or variability of these dynamics. It is also important to use more comprehensive measures of jealousy that extend beyond digital contexts, capturing a wider array of jealousy experiences. Moreover, forthcoming studies should aim for more balanced samples across sexes as well as more diverse samples that include various relationship types, allowing for meaningful comparisons across these groups. Power analyses to justify sample sizes and ensure robust and generalizable findings should also be incorporated. Finally, the non-pathological and subtle nature of normal relationship jealousy that was studied should not be confused with other kinds of destructive relationship affects and behaviors within romantic relationships (i.e., [61]), which were not the focus of the research.

## 5. Conclusions

This study reaffirms the idea that love and jealousy may be evolved protective factors of romantic relationships. It stresses the complex, culturally nuanced nature of jealousy in romantic relationships. The findings emphasize the interplay of attachment, relational closeness, and feeling loved, highlighting the need to consider both individual and cultural factors in understanding emotional experiences within romantic partnerships.

## Figures and Tables

**Figure 1 behavsci-15-00132-f001:**
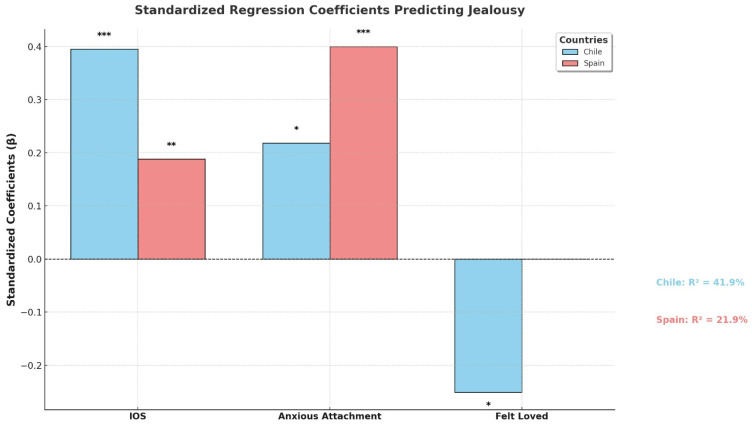
Predictors of jealousy by country. Note: * *p* < 0.05, ** *p* < 0.01, and *** *p* < 0.001.

**Table 1 behavsci-15-00132-t001:** Descriptive statistics and effect sizes for the variables by country.

	Chile		Spain		*d*	*p*
	*M*	*SD*	*M*	*SD*		
Jealousy	2.48	1.27	1.98	0.91	0.47	0.001
Intimacy	22.27	3.03	21.80	2.84	0.16	0.208
Passion	19.87	4.17	19.00	3.90	0.22	0.090
Commitment	22.03	3.72	21.94	2.94	0.03	0.828
IOS	3.74	4.00	3.73	4.00	0.01	0.954
Felt Loved	6.24	1.04	6.10	0.95	0.15	0.253
Anxious Attachment	2.37	1.07	2.37	0.83	0.00	0.997
Avoidant Attachment	2.34	0.64	2.42	0.58	0.12	0.340

Note: n = 265. Values of 0.20, 0.50, and 0.80 for Cohen’s d (*d)* are commonly considered to be indicative of small, medium, and large effects, respectively.

**Table 2 behavsci-15-00132-t002:** Correlation between jealousy and the affective variables.

Variables	2	3	4	5	6	7	8
1. Jealousy	−0.127 *	0.076	−0.173 *	0.364 ***	−0.240 ***	0.430 ***	0.178 **
2. Intimacy		0.667 ***	0.708 ***	0.013	0.815 ***	−0.174 **	−0.175 **
3. Passion			0.587 ***	0.095	0.577 ***	0.019	−0.042
4. Commitment				0.036	0.677 ***	−0.194 **	−0.122
5. IOS					−0.054	−0.236 **	−0.169 **
6. Felt loved						0.243 **	0.071
7. Anxious Att.							0.445 ***
8. Avoidant Att.							-----

Note. The table presents Pearson correlation coefficients (r) between jealousy and the affective variables: love (including the dimensions intimacy, passion, and commitment), closeness (assessed using IOS), felt loved, and attachment (anxious attachment and avoidant attachment). * *p* < 0.05, ** *p* < 0.01, and *** *p* < 0.001.

## Data Availability

Data analyzed during this study are not publicly available due to confidentiality agreements with participants but are available from the corresponding author upon reasonable request.

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
