# Peer review of "What Is the Link of Closeness and Jealousy in Romantic Relationships?"

_behavsci, 2025, doi:10.3390/bs15020132_

Round 1

Reviewer 1 Report

Comments and Suggestions for Authors

The present study explored the relationship between jealousy and affective components of love, and produced findings that would be of interest to researchers working in the field. I have some suggestions for minor corrections that I summarize below:

Introduction

In order to increase readability, you could consider dividing the introduction into subsections.

Participants

Provide the N for men and women, separately for each country.

Measures

Consider providing examples of items for each measure used.

Data analysis

Consider expanding this section to include more details about the statistical analysis used.

Discussion

Report as a limitations that your sample was not balanced across sexes.

Author Response

Reviewer 1

The present study explored the relationship between jealousy and affective components of love, and produced findings that would be of interest to researchers working in the field. I have some suggestions for minor corrections that I summarize below:

Introduction

In order to increase readability, you could consider dividing the introduction into subsections.

R: Thank you for your suggestion. In the revised manuscript, we have divided the introduction into three subsections:

1.1. Love and attachment 

1.2. Jealousy 

1.3. Relationship between love and jealousy 

Participants

Provide the N for men and women, separately for each country.

R: Thank you for pointing out this issue. We have added this information in the participants section of the manuscript.   

Measures

Consider providing examples of items for each measure used.

R: Many thanks. We have added example items for two of the scales (The Digital Jealousy Scale and The Revised Adult Attachment Scale) to improve clarity. 

Example items for the remaining three scales employed in our study (TLS-15, Felt-Loved and IOS scales) were already included in the original draft. 

Data analysis

Consider expanding this section to include more details about the statistical analysis used.

R: Thank you for your feedback. We have expanded this section to provide additional details regarding the statistical methods used.

Discussion

Report as a limitations that your sample was not balanced across sexes.

R: Thank you. We have now included a statement in the discussion addressing this issue.

Reviewer 2 Report

Comments and Suggestions for Authors

This study provides an interesting perspective on the relationship between jealousy and affective components of love. However, some aspects need to be revised in order to improve the completeness of the manuscript.

1. the abstract is well written, but it is recommended to add a description of the research tool.

2. The background of the study has already been thoroughly described in the Introduction section. However, given that this study is a comparative cultural study in two Spanish-speaking countries in South America and Europe, I would suggest that further literature on cultural differences be added to the Introduction section.

3. The manuscript mentions that the data were collected through an online platform, but does not provide specific information about the platform, the criteria for sample selection, and potential biases in the data collection process.

4. How did the authors justify the sample size? What is power analysis findings?

5. Table 1. Typography needs to be aligned, what does ***p < .00 want to show?

6. Symbols such as M, SD, p and t need to be italicized, please check the normality of the results section.

7. The results of this study are presented in Tables I and II, and it is recommended that some important results be visualized in the form of figure.

8. In the discussion section, the data results are mainly interpreted, and there is almost no citation of previous similar or opposite research literature. It is suggested that some classic literature be cited from the cultural point of view in conjunction with the results of this study to reopen the discussion.

 Author Response

This study provides an interesting perspective on the relationship between jealousy and affective components of love. However, some aspects need to be revised in order to improve the completeness of the manuscript.

  1. the abstract is well written, but it is recommended to add a description of the research tool.

We added more detail on the research tool on the abstract, specifying we used the digital jealousy scale, and the opposite nature of closeness and jealousy which inspired the study.

  1. The background of the study has already been thoroughly described in the Introduction section. However, given that this study is a comparative cultural study in two Spanish-speaking countries in South America and Europe, I would suggest that further literature on cultural differences be added to the Introduction section.

Based on the reviewer suggestion, we have added more cultural aspects and details regarding the measurement of both emotions to enhance the understanding of the manuscript. We believe these additions improve the context and depth of the study.

  1. The manuscript mentions that the data were collected through an online platform, but does not provide specific information about the platform, the criteria for sample selection, and potential biases in the data collection process.

We thank you for this comment, and have added now that we run the study using Psytoolkit platform. We also clarified that we used a convenience sample, which is also discussed as a potential bias in data collection.

  1. How did the authors justify the sample size? What is power analysis findings?

We appreciate your observation. The sample size for this study was determined based on the availability of participants who met the inclusion criteria (being over 18 years old, currently in a romantic relationship, and residing in Chile or Spain) rather than through an a priori power analysis. Nevertheless, with 265 participants, we ensured a sufficient sample size for conducting the planned analyses. We acknowledge that incorporating a formal power analysis could strengthen the methodological rigor of future studies, and we have noted this as a limitation in the revised manuscript.

  1. Table 1. Typography needs to be aligned, what does “***p < .00” want to show?

We have aligned Table 1, and we specified the meaning of “***p < .00”

  1. Symbols such as M, SD, p and t need to be italicized, please check the normality of the results section.

Thanks, we have italicized the statistical symbols and added normality check on the results section.

  1. The results of this study are presented in Tables I and II, and it is recommended that some important results be visualized in the form of figure.

We have added a Figure with the results of the standardized predictors of Jealousy by country, to better convey the visualization of our results.

  1. In the discussion section, the data results are mainly interpreted, and there is almost no citation of previous similar or opposite research literature. It is suggested that some classic literature be cited from the cultural point of view in conjunction with the results of this study to reopen the discussion.

We agree on the interpretative nature of our original discussion. So now we have developed the discussion section to dialogue with the research literature we provided and a couple of studies that may help interpret our findings.

Reviewer 3 Report

Comments and Suggestions for Authors

Dear Editor:

This paper deals with an important topic (romantic love and jealousy) and the estudy is carried out with samples from two countries (Spain and Chile), but it has serious limitations for it to be published.

1) The theorical introduction should be improved. It should define the concept of jealousy, better specify the evolutionary theoretical perspective and briefly explain the differences between men and women when experienceing jealousy, as well as especify the research and some lines of research carried out in the field of social psychology and psychopatology.

2) The objetive and hypothesis should be better explained. They should be more explicit in the theoretical introduction.

3) They should explain the cultural differences in the variables used in this study in both countries.

4) They should make up the characteristics of the samples from the two countries more explicit with regard to the socio-demographic variables.

5) The authors should improve statistical analysis.

6) They must revise the references.

Best regards

Author Response

This paper deals with an important topic (romantic love and jealousy) and the estudy is carried out with samples from two countries (Spain and Chile), but it has serious limitations for it to be published.

1) The theorical introduction should be improved. It should define the concept of jealousy, better specify the evolutionary theoretical perspective and briefly explain the differences between men and women when experienceing jealousy, as well as especify the research and some lines of research carried out in the field of social psychology and psychopatology.

Based on your suggestion, we have defined the concept of jealousy, better specify the evolutionary theoretical perspective, and incorporated studies on gender differences in jealousy from an evolutionary perspective in the context of infidelity scenarios, which further enriches the discussion and aligns with your recommendations.

2) The objetive and hypothesis should be better explained. They should be more explicit in the theoretical introduction.

The last paragraph of the introduction already included our hypotheses and objectives. However, we have revised this section to make the objectives and hypotheses more explicit and to strengthen their connection to the theoretical framework, which has been expanded throughout the introduction.

3) They should explain the cultural differences in the variables used in this study in both countries.

We have extended on cultural differences, with recent literature on attachment and jealousy in the variables used in the study, in the introduction as well as in the discussion, in order to enrich the scope of our results considering the limitations of our design.

4) They should make up the characteristics of the samples from the two countries more explicit with regard to the socio-demographic variables.

We have extended on socio-demographic variables more explicit at the beginning of the results.

5) The authors should improve statistical analysis.

We have tried to improve the statistical analysis by adding more details on sample composition, limitations and the like. 

6) They must revise the references.

We thank you for pointing this, and we have improved the reference format in this version.